# What Is the Impact of Unemployment as an Adverse Experience? Post-Traumatic Stress Disorder and Complex Post-Traumatic Stress Disorder: A Meta-Analysis

**DOI:** 10.3390/ijerph22050696

**Published:** 2025-04-28

**Authors:** Marcelo Nvo-Fernandez, Valentina Miño-Reyes, Carlos Serrano, Hedy Acosta-Antognoni, Fabiola Salas, Claudio Vásquez Wiedeman, Francisco Ahumada-Méndez, Marcelo Leiva-Bianchi

**Affiliations:** 1Laboratory of Methodology, Behavioural Sciences and Neuroscience, Faculty of Psychology, Universidad de Talca, Talca 3460000, Chile; navarro.marcelo.f@gmail.com (M.N.-F.); valentina.m.reyes@gmail.com (V.M.-R.); fabiolasalassalas@gmail.com (F.S.); francisco.ahumada@utalca.cl (F.A.-M.); marcleiva@utalca.cl (M.L.-B.); 2Faculty of Social Sciences and Humanities, Universidad Autónoma de Chile, Talca 3460000, Chile; carlosserrano159@gmail.com; 3Research Team on Organizational Psychology, Faculty of Psychology, Universidad de Talca, Talca 3460000, Chile; hacosta@utalca.cl

**Keywords:** unemployment, PTSD, CPTSD, risk factors, meta-analysis

## Abstract

This meta-analysis examined how unemployment, a psychosocial stressor, influences post-traumatic stress disorder (PTSD) and complex PTSD (CPTSD). It specifically explores unemployment as a risk factor for trauma, with emphasis on CPTSD, and investigates economic variables, including the GINI coefficient, as potential moderators. A systematic search in Web of Science, Scopus, and PubMed yielded 33 studies comprising more than 57,000 participants. Odds ratios (OR) were computed, and a random-effects model was used to synthesize the findings. Meta-regression analyses were conducted to evaluate the effects of economic inequality (GINI) and nominal gross domestic product (NGDP), but neither moderator reached statistical significance; this is addressed in detail in the Discussion. The results revealed that unemployment significantly elevated the risk for PTSD (OR = 1.500; logOR = 0.3826; PI: 0.457–4.702) and CPTSD (OR = 2.180; logOR = 0.7430; PI: 0.501–8.808), with a stronger impact on CPTSD. These findings emphasize unemployment as a pivotal predictor of trauma, especially CPTSD, broadening the traditional focus on interpersonal factors. They also highlight the importance of integrating psychosocial and economic variables into clinical assessments and public health policies. Addressing both unemployment and economic inequality could be critical for effective interventions and prevention efforts, underscoring the need for further multidisciplinary research.

## 1. Introduction

Employment not only serves as a cornerstone for economic subsistence but also plays a critical role in shaping individuals’ social, psychological, and community well-being [1,2]. Consequently, job loss has far-reaching implications, affecting the physical, psychological, and social domains [3,4,5]. Despite continuous changes in the global economy, unemployment consistently remains above 5% worldwide, reaching nearly 7% during the 2020–2022 pandemic, according to World Bank data (2023) [6]. This rise in unemployment rates has been closely linked to a significant deterioration in mental health, increasing the risk of conditions such as anxiety, depression, and other psychiatric disorders [7,8,9].

The psychosocial impact of unemployment is well-documented. Previous studies have shown that job loss significantly increases psychological distress among affected individuals. Notably, Griffiths et al. reported that unemployed individuals with limited financial resources are at a heightened risk of experiencing severe psychological distress, with an adjusted odds ratio (AOR) as high as 8.36 (CI: 3.35–20.87) [9,10]. This finding underscores the relationships between unemployment, economic constraints, and mental well-being, highlighting the magnitude of the impact on vulnerable groups. Moreover, mental health issues not only emerge as consequences of unemployment but also act as barriers to reemployment, perpetuating a cycle of poor mental health and unemployment [11,12]. Prolonged unemployment exacerbates vulnerability, with depression rates reaching 50% among individuals who have been unemployed for over 12 months [13]. However, reemployment, even after long periods of inactivity, can lead to significant improvements in overall mental health [14,15].

Despite extensive research on the impact of unemployment on mental health, the relationship between unemployment and trauma has received relatively less attention. While most trauma studies focus on adverse childhood experiences or exposures in war contexts [16,17], recent findings suggest that unemployment in adulthood can also serve as a significant traumatic stressor. For instance, unemployment has been shown to double the likelihood of developing post-traumatic stress disorder (PTSD) [18]. Some theoretical models propose a linear decline in mental health as the duration of unemployment increases, while others suggest adaptive processes that stabilize mental health at low levels over time [5,6,7].

It is important to acknowledge that standard diagnostic criteria for PTSD and complex PTSD (CPTSD) traditionally require exposure to a high-intensity traumatic event—one that involves a direct threat to life or physical integrity [19]. However, this study explores the hypothesis that chronic adverse experiences, such as prolonged unemployment, can also induce trauma-related symptoms. Emerging evidence and theoretical work suggest that sustained stress, social isolation, and economic insecurity associated with job loss may elicit psychological responses (e.g., re-experiencing, avoidance, persistent sense of threat) comparable to those seen in classic trauma exposures. This perspective is supported by empirical findings linking unemployment to heightened traumatic stress [18] and by conceptual models that advocate for an expanded definition of trauma to encompass chronic, non–life-threatening adversities.

Unemployment can act as a catalyst for both simple and complex trauma, particularly in high-stress environments, such as post-disaster contexts [20]. Individuals with PTSD often face challenges in maintaining employment due to debilitating symptoms, such as severe anxiety and social impairments, creating a negative feedback loop between psychological distress and joblessness [21]. Furthermore, rapid technological advancements and the increasing integration of artificial intelligence into labor markets exacerbate these concerns. Fears of job displacement generate stress and anxiety, while excessive reliance on technology and social media can intensify feelings of isolation and worsen mental health problems [22]. These effects are even more pronounced in contexts with high socioeconomic inequality, where limited community resilience magnifies the mental health impact of unemployment [23,24].

In this context, a pertinent question arises: Can unemployment generate symptoms characteristic of complex post-traumatic stress disorder (CPTSD), such as emotional dysregulation, a negative self-concept, and difficulties in establishing relationships with others? Since 2018, the World Health Organization, through the ICD-11 manual, has established that a CPTSD diagnosis requires the presence of three primary symptom clusters: (1) re-experiencing the traumatic event, (2) avoidance of trauma-related memories, and (3) a persistent sense of threat. While CPTSD shares these core symptoms with PTSD, the ICD-11 defines it as a condition that also includes three additional domains specific to disturbances in self-organization (DSO): affective dysregulation, a persistently negative self-concept, and significant interpersonal difficulties. These DSO domains are exclusive to CPTSD, underscoring its clinical complexity.

Understanding this study within a psychosocial context is essential. For this reason, the choice of the GINI index and nominal GDP as moderating variables reflects the need to capture socioeconomic dimensions influencing the experience of unemployment as a stressor. The GINI index, by reflecting income distribution inequality, is associated with access to economic and social resources [23,24,25]. In contexts with lower inequality, greater access to social support and mental health services mitigates the negative effects of unemployment. Conversely, in societies with high inequality, gaps in social safety nets tend to exacerbate psychological vulnerability and amplify the impact of unemployment [26]. However, some studies suggest that extreme levels of inequality may lead to a “normalization” of socioeconomic imbalances, attenuating the perception of unemployment as traumatic in certain groups [27].

Nominal GDP, in turn, provides an approximation of a country’s overall economic development and the strength of its social protection systems. A higher GDP is often associated with robust employment programs, broader healthcare coverage, and better working conditions, which may reduce the impact of unemployment on mental health [28]. In contrast, low-GDP contexts show resource limitations and greater economic instability, intensifying the psychological burden of unemployment [25]. Therefore, a combined analysis of the GINI index and nominal GDP allows for an evaluation of both the magnitude of national wealth and its distribution, offering a comprehensive perspective on how structural and cultural conditions modulate the relationship between unemployment and the development of PTSD/CPTSD.

Therefore, this study evaluated unemployment status as a risk factor (adverse experience) for the development of PTSD and CPTSD, while also specifically exploring the moderating roles of the GINI index (a measure of inequality) and nominal GDP. Based on the ICD-11 framework, this research analyzed how unemployment, as a sustained stressor, might influence the emergence of distinctive PTSD symptoms. By integrating broader socioeconomic factors, this work aimed to contribute to a deeper and more nuanced understanding of how labor market dynamics and trauma interact within an ever-evolving global economic landscape.

## 2. Search Strategy

The reporting methods and procedures were carried out in accordance with the PRISMA checklist (Preferred Reporting Items for Systematic Reviews and Meta-Analyses) [29,30]. Articles in English published in peer-reviewed journals up to November 2024 were included. The search was conducted in three scientific literature databases: Web of Science, Scopus, and PubMed. The search terms used in these bibliographic databases included the following combinations: TITLE (“complex PTSD” OR “complex posttraumatic stress disorder” OR “CPTSD” OR “PTSD” OR “posttraumatic stress disorder”) AND TITLE (“risk factors” OR “predictors” OR “unemployment”) (Table 1). Given the quantitative focus of this study, documents such as book chapters, theoretical reviews, systematic reviews, editorial comments, letters or notes, case studies, and other articles that did not provide quantitative information on risk factors for CPTSD were excluded.

## 3. Inclusion and Exclusion Criteria

The studies included in the meta-analysis met the following criteria. First, they examined unemployment as a possible risk factor (predictor) for the development of PTSD and/or CPTSD. Second, they reported at least one of the following data points: (1) Odds ratio (OR) and corresponding 95% confidence intervals (CI) or (2) the frequency of unemployment as a risk factor in the population with PTSD and/or CPTSD and in the population without PTSD and/or CPTSD, from which a data conversion was carried out for subsequent analysis.

Studies were excluded if they addressed physical traumas (specifically musculoskeletal and/or neurological pathologies) and not PTSD and/or CPTSD; if they did not present meta-analyzable data (OR or frequencies); or if they contained insufficient data to calculate univariate effect sizes and it was not possible to obtain such information from the study’s author. Likewise, any studies that did not include unemployment as a risk factor or examined unemployment experienced by someone other than the study sample (e.g., a relative) were excluded, as were reviews or qualitative studies that did not present new data or only included qualitative analyses. Further exclusions involved single-case designs. Finally, if more than one article presented data from the same sample, the most recent and comprehensive article was included in the meta-analysis.

## 4. Quality Assessment

The quality assessment process was carried out using the Quality Assessment Tool for Observational Cohort and Cross-Sectional Studies (National Institute of Health). This tool comprises 14 items that evaluate aspects such as the formulation of the research question, the composition of the population, recruitment procedures, sample size, methods for measuring exposures and outcomes, attrition rates, and the statistical methods used. Each item is rated as “yes”, “no”, or partial (“cannot determine”, “not applicable”, or “not reported”). Following the recommendations of Katy Gaythorpe and colleagues [30], a score of 1.0 was assigned to “yes” responses, 0.5 to partial responses, and 0 to “no” responses. Based on the total score, studies were classified as high quality when they scored 7.0 points or higher, moderate quality when they scored between 5.0 and 6.0 points, and low quality when they scored below 5.0 points (Appendix A).

## 5. Data Extraction

The initial search was performed by one of the authors, who identified articles based on the established search terms. A subsequent manual review was carried out to select articles that met the inclusion criteria. This review was conducted independently by three authors, who examined the titles and abstracts, or, when necessary, the full text of relevant articles. Discrepancies in interpretation were resolved through discussion with an additional reviewer, reaching a final consensus for selection and coding.

The selected articles were thoroughly reviewed in accordance with the inclusion criteria. For classification purposes, a color-coding system was used based on Cochrane’s recommendations [31]: included articles were marked in green, excluding articles in red, those with uncertainties in yellow, and those not found in orange. Several tools were used for data extraction and management, including Microsoft Excel 2021 to organize and compile data, Mendeley Reference Manager (version 2.88) to manage the articles (selection, eligibility, and inclusion) [31,32], and Review Manager (RevMan) (version 5.4.1) for analysis and reporting of results.

From each eligible study, the following data were extracted: the first author’s last name, year of publication, study location, sample size, percentage of female participation, participant ages and standard deviations, prevalence of CPTSD, number of individuals exposed to potentially traumatic factors, estimated effect size (OR) with corresponding 95% confidence intervals (CI), adjusted covariates in the statistical analysis, and the frequency of exposure to the risk factor. Additionally, the variance of the logarithm of each effect size was calculated for the analysis.

## 6. Statistical Analysis

Odds ratios (OR) were used as the primary measure of effect size in this analysis. ORs were extracted directly from the studies whenever they were reported. In cases where OR values were not provided, they were calculated from the reported exposure frequencies. Additionally, other effect size measures, such as correlations between risk factors and PTSD and/or CPTSD, as well as mean differences between exposed and unexposed groups, were converted to ORs to ensure methodological consistency. The variance of the natural logarithm of the OR (logOR) was calculated using rESCMA, an open-access web-based calculator for effect size conversions (https://www.rescma.com/) (accessed on 26 March 2025) [32].

A random-effects model was adopted as the analytical framework under the assumption that true effect sizes vary among the included studies [33]. To stabilize variances and ensure uniform distributions, ORs were transformed into natural logarithms (logORs), and their variances (VarLogORs) were calculated prior to analysis. The results were then back-transformed to ORs to facilitate interpretation [33,34]. Publication bias was assessed using funnel plots and Egger’s regression test. Symmetry in the funnel plots indicates the absence of bias, whereas asymmetry suggests the possibility of publication bias. The analyses were conducted in RStudio version 2023.06.0 and R version 4.3.1, using the metafor package [35].

To explore the impact of socioeconomic variables on the relationship between unemployment and the development of PTSD and/or CPTSD, a meta-regression analysis was performed using mixed-effects models, which consider both within-study and between-study variability. The selected variables included nominal Gross Domestic Product (GDP) as an indicator of inflation-adjusted economic growth and the Gini coefficient (GINI) as a measure of income inequality [36], with data extracted from the World Bank databases [6]. The significance of the moderators was evaluated using β values (*p* < 0.05), classified as low (<24%), moderate (25–64%), or high (>65%) [37]. This analysis provided deeper insight into how these moderating factors influence the overall effect estimates.

In both the meta-analyses and meta-regressions, Cochran’s Q test and the I^2^ index were used to evaluate heterogeneity. A significant *p*-value in the Q statistic indicates heterogeneity among studies, while the I^2^ index quantifies the percentage of variation attributable to this heterogeneity. I^2^ values were interpreted according to established thresholds: low (<40%), moderate (40–60%), and substantial (>60%). The results of the meta-analyses were presented through forest plots, which visually compare individual effect sizes and the combined effect [38]. Additionally, publication bias was evaluated using Egger’s test (*p* < 0.05). This systematic approach ensures that the conclusions drawn are robust and reliable in the face of potential biases or external influences.

## 7. Results

A total of 1427 records were identified in the Web of Science, SCOPUS, and PubMed databases. Of these, 33 studies were included (Figure 1), representing 57,951 individuals from 21 countries and various populations, including general, clinical, and trauma-exposed groups (Table 2). Regarding study quality, the average score was 6.83, indicating an overall assessment of study quality as fair (Table 3, Appendix A).

## 8. Quality Assessment

The methodological quality of the included studies varied. Table 2 provides a summary of the 14 quality assessment areas and the general rating for each study, indicating the potential risk of bias. According to the guidelines provided by the National Institutes of Health tool, out of the 33 studies analyzed, 21 studies were rated as “good” quality, suggesting a low risk of bias, while 12 studies received a “moderate” quality rating, indicating a moderate risk of bias.

In this research, a significant number of the included studies are cross-sectional in design. It is important to note that cross-sectional designs have inherent limitations in assessing changes over time, as they capture information at a single point in time. This explains the minimal scores on questions pertaining to characteristics typical of longitudinal studies: sufficient time to observe associations (Item Q7), whether exposures were evaluated more than once over time (Item Q10), and assessment of participant loss during follow-up (Item Q13).

## 9. Meta-Analysis

Among the 33 studies included in this meta-analysis, unemployment was evaluated as a risk factor for the development of PTSD and CPTSD, integrating both combined odds ratio (OR) estimates and the exploration of socioeconomic moderator variables (e.g., nominal GDP and the GINI coefficient). Below, the main findings of each meta-analysis are presented, highlighting their heterogeneity, potential publication bias, and the robustness of the associations.

For PTSD, 31 studies were analyzed, encompassing a total of *n* = 56,905 participants. This combined sample offers a broad perspective across various contexts (general populations, clinical settings, and trauma-exposed groups) and types of research designs (cross-sectional and longitudinal). The odds ratio (OR) obtained was 1.500 (logOR = 0.3826, *p* < 0.000; 95% CI: 0.165–0.600), suggesting that unemployment is associated with an approximately 50% increase in the likelihood of developing PTSD (Table 4).

However, the prediction interval (0.457–4.702) revealed considerable variation among the studies, indicating that the effect may fluctuate according to factors such as duration of unemployment, resilience of social support networks, and specific sociocultural environments. Moreover, substantial heterogeneity was observed (Q = 804.155, *p* < 0.000; I^2^ = 98.02), reflecting notable differences in study designs and populations. This dispersion could be explained by variability in diagnostic criteria, sample sizes, or the combination of cross-sectional and longitudinal studies. Finally, Egger’s test (Z = 2.0813, *p* = 0.037) indicated potential publication bias, necessitating caution when interpreting the robustness of this finding (Figure 2).

For CPTSD, 12 studies were included (*n* = 8047), focusing on diverse populations (refugees, general communities, and clinical groups). A combined OR of 2.180 was found (logOR = 0.743, *p* < 0.000; 95% CI: 0.346–1.138), indicating that unemployment may double the likelihood of developing CPTSD. The prediction interval (0.501–8.808) was even broader than for PTSD, underscoring the influence of specific contextual factors and the high variability in the manifestation of complex symptoms.

The heterogeneity analysis showed very high levels (Q = 405.014, *p* = 0.000; I^2^ = 96.42), suggesting significant differences among the studies in terms of methodology, characteristics of the populations, and possible clinical cofactors (e.g., previous traumas or concurrent mental health conditions). Similarly, Egger’s test (Z = −0.2670, *p* = 0.789) did not detect publication bias (Figure 3).

The meta-analyses of PTSD and CPTSD reinforce the significant association between unemployment and the risk of developing a post-traumatic stress disorder, including a more complex form. Despite recognizing high heterogeneity and the presence of publication bias, the overall estimates (OR of 1.500 and 2.180, respectively) demonstrate that unemployment acts as a stressor with the potential to exacerbate traumatic symptoms.

Table 5 presents the results of the meta-regression aimed at assessing the impact of two economic variables, the Gini index and nominal gross domestic product (nominal GDP, NGDP), on the relationship between unemployment and the onset of PTSD and CPTSD. The interpretation of each moderation model is described below, considering the estimator values, their confidence intervals, and statistical significance levels.

First, the estimated value for the Gini index (estimate = −0.0242, SE = 0.0221, *p* = 0.2733, CI: −0.0659 to 0.0224) did not reach statistical significance. This suggests that economic inequality, as measured by the Gini index, does not conclusively alter the impact of unemployment on the development of PTSD across the studies analyzed.

Similarly, nominal GDP (NGDP) also did not exhibit a significant moderating effect (estimate = −0.0000, SE = 0.0000, *p* = 0.4926, CI: −0.0000 to 0.0000). In both cases, the heterogeneity tests (Q = 799.9109, *p* < 0.001, I^2^ = 97.76% for Gini; Q = 750.4517, *p* < 0.001, I^2^ = 97.83% for NGDP) revealed substantial variability among the studies. This very high level of heterogeneity suggests the presence of methodological, clinical, or contextual factors not captured by the meta-regression models.

Regarding the Gini index, the estimation (estimate = −0.2125, SE = 0.1014, *p* = 0.0649, CI: −0.4113 to −0.0136) produced a *p*-value above the conventional threshold of *p* < 0.05 and thus is not considered statistically significant. Although the confidence interval mostly lies below zero, which could imply a tendency toward a negative moderating effect, the *p*-value does not confirm this moderation conclusively.

As for nominal GDP (estimate = −0.0000, SE = 0.0000, *p* = 0.682, CI: −0.0000 to 0.0000), no statistically significant moderating effect was observed on the unemployment–CPTSD relationship. Similar to the findings for PTSD, heterogeneity was again very high (Q = 223.8095, *p* < 0.001, I^2^ = 94.94% for Gini; Q = 404.1999, *p* < 0.001, I^2^ = 96.58% for NGDP). Overall, the results show that neither of the two moderators (Gini nor NGDP) reached statistical significance (*p* < 0.05) in modifying the association between unemployment and the onset of PTSD or CPTSD.

## 10. Discussion

This meta-analysis had two main objectives: (1) to quantify the association between unemployment and the risk of developing both PTSD and CPTSD, and (2) to evaluate whether economic inequality (measured by the Gini index) and nominal GDP act as possible moderators of this relationship. By including 31 studies with 56,905 participants for PTSD and 12 studies with 8047 participants for CPTSD, a solid foundation was established for examining these objectives.

Regarding the first objective, the findings indicate that unemployed individuals face a significantly higher risk of developing PTSD—approximately 50% higher—compared to those who are employed. Similarly, the likelihood of developing CPTSD doubles among unemployed individuals relative to their employed counterparts. This increase in the risk of CPTSD underscores the potential magnitude of the psychosocial impact of unemployment, likely tied to the loss of structure, purpose, and support networks often accompanying joblessness. Our results, consistent with previous analyses [20], strengthen the notion that prolonged unemployment can serve as a precipitating factor for chronic trauma.

Scientific literature on CPTSD has documented clear differences from PTSD, particularly in the realms of emotional and behavioral disturbances, negative self-perception, and significant interpersonal difficulties [54,70]. Our results suggest that these disturbances may be aggravated in unemployed individuals, supporting the hypothesis of a closer link between unemployment and CPTSD than between unemployment and PTSD. In this vein, chronic lack of employment emerges as a source of trauma that can be not just acute but also ongoing, influencing self-esteem, social relationships, and psychological resources [12,13]. These findings underscore the importance of distinguishing between PTSD and CPTSD, as each involves unique psychological and psychosocial processes.

The psychosocial impact of a trauma model applied to unemployment [71,72] provides additional insight: beliefs about oneself, work, personal merit, and social justice affect the psychological response to job loss. While some individuals develop resilience, seeking new employment, strengthening social networks, or engaging in activities that restore their sense of efficacy, others may experience PTSD or CPTSD symptoms, even when protective factors such as government or family support are present. These individual differences explain why, in seemingly similar contexts (e.g., countries with robust welfare policies vs. those with weaker systems), divergent responses are observed.

Another point of interest lies in the relationship between unemployment, social stigmatization, and self-esteem [73]. The stigma associated with being unemployed may intensify among those who have recently lost their jobs or who have been unable to find work for an extended period, negatively affecting perceptions of self-worth [74]. Unlike more conventional PTSD risk factors, such as war or natural disasters that typically involve acute and life-threatening experiences, unemployment can represent a chronic, ongoing stressor marked by economic insecurity, social isolation, and persistent uncertainty. This sustained nature of the stress may more readily contribute to difficulties in emotional regulation, negative self-concept, and relational conflicts—core features of CPTSD [20,75]. Consequently, individuals suffering from unemployment-related stress may require clinical interventions that extend beyond short-term trauma processing and include strategies for building resilience, social support, and economic or vocational assistance (e.g., job placement programs, skills training, or financial counseling). Such approaches can address both the psychological and practical aspects of a stressor that is often long-lasting and structurally embedded. At the same time, the so-called Fourth Industrial Revolution brings new challenges for mental health and employment, such as techno-anxiety and techno-stress. 

With respect to the second objective, this study explored whether the Gini index and nominal GDP moderate the association between unemployment and the development of PTSD/CPTSD. Although previous literature has suggested that economic inequality may influence various mental health conditions [26,27], our analyses did not yield statistically significant results for either of these indicators. In other words, neither income inequality (Gini) nor nominal GDP showed a conclusive moderating effect on the unemployment–PTSD or unemployment–CPTSD relationship. These findings partially contrast with certain theoretical studies suggesting that, in contexts of high inequity, a “normalization” of precarious employment may exist, reducing the guilt or shame associated with job loss.

However, the *p*-values from our meta-regressions do not support a statistically robust moderation. This does not rule out, from a conceptual standpoint, the relevance of exploring how unemployment is perceived in environments with greater or lesser socioeconomic gaps. In societies with high levels of inequality, chronic exposure to shortages and lower expectations for job mobility could theoretically alter the way unemployment is experienced [76]. Moreover, the absence of formal protection could be offset by informal community or family support networks [77], partially mitigating the psychological impact of joblessness [23].

Additionally, while our meta-analysis emphasized risk factors such as unemployment, several of the included studies briefly touched upon social and psychological protective factors. For instance, a few studies reported that individuals with stronger support networks, access to mental health services, or governmental aid programs showed lower levels of post-traumatic symptoms [16,78,79,80,81,82,83]. However, none of the studies systematically tested these constructs as formal moderators in the unemployment–PTSD/CPTSD relationship. This gap indicates a broader need to explore how resilience and coping mechanisms, including community-based mental health resources, may buffer the effects of job loss. Future research should adopt designs that explicitly assess these variables, potentially elucidating why certain unemployed individuals do not develop clinically significant trauma symptoms and thereby offering a more balanced perspective that accounts for both risk and protective factors.

From an applied perspective, the conclusions of this meta-analysis have implications for public policies and health interventions. Recognizing unemployment as a risk factor for post-traumatic stress disorders, particularly in their complex form, could guide prevention strategies and care that focus on vulnerable groups. Additionally, it is essential to encourage lines of research examining the interaction of cultural, cognitive, and social variables—beyond purely economic ones—that may influence the subjective experience of unemployment.

Nevertheless, this study is not without limitations. The high clinical and statistical heterogeneity indicates that the results should be interpreted cautiously, and the lack of longitudinal studies makes it difficult to establish clear causal relationships. Furthermore, factors such as gender, age, family situation, the specific duration of unemployment, or the type of employment are not always reported in a standardized manner in the literature, preventing more detailed analysis. Lastly, publication bias and wide confidence intervals highlight the need to increase the number of studies with large, representative samples following rigorous protocols.

Overall, the findings of this meta-analysis underscore the urgency of considering unemployment as a trauma trigger capable of severely affecting mental health. Although no significant moderating effect of the Gini index or nominal GDP was observed, understanding the psychosocial foundations of unemployment and its relationship to the development of PTSD and CPTSD offers a starting point for designing prevention strategies more attuned to the realities of unemployed individuals. Ultimately, the confluence of labor, social, economic, and cultural factors emphasize the complexity of the relationship between unemployment and trauma, and points to the need for further research exploring the nuances of this interaction.

## Figures and Tables

**Figure 1 ijerph-22-00696-f001:**
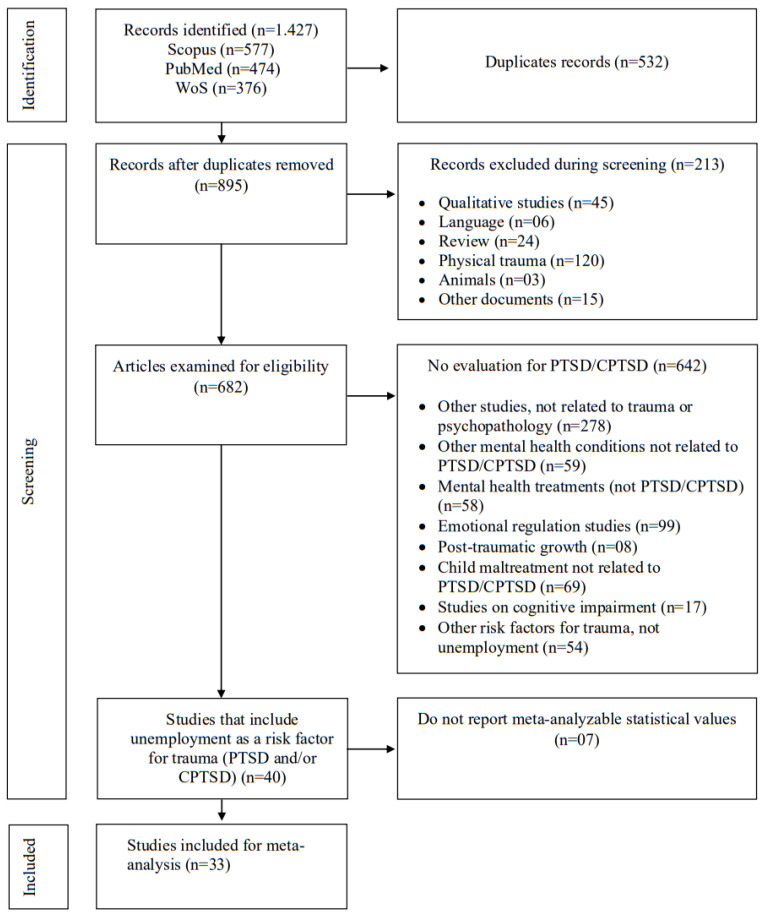
Flow diagram.

**Figure 2 ijerph-22-00696-f002:**
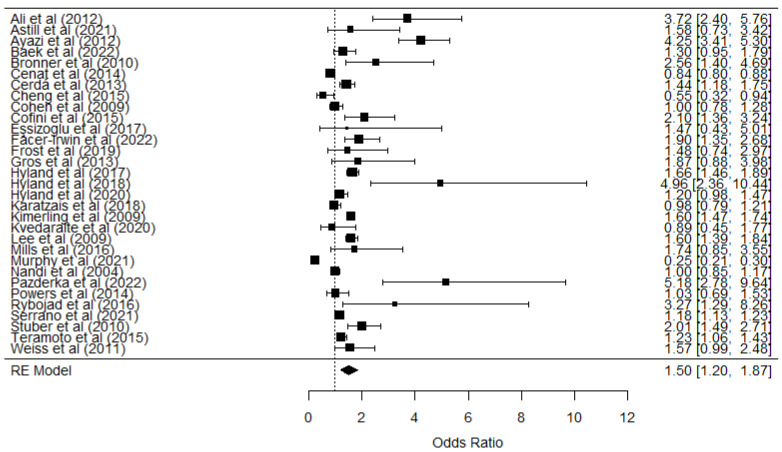
Forest plot of the unemployment and PTSD relationship [17,21,39,40,41,42,43,44,45,46,47,48,49,50,51,52,53,54,55,56,57,58,59,60,61,62,63,64,65,66,67,68,69].

**Figure 3 ijerph-22-00696-f003:**
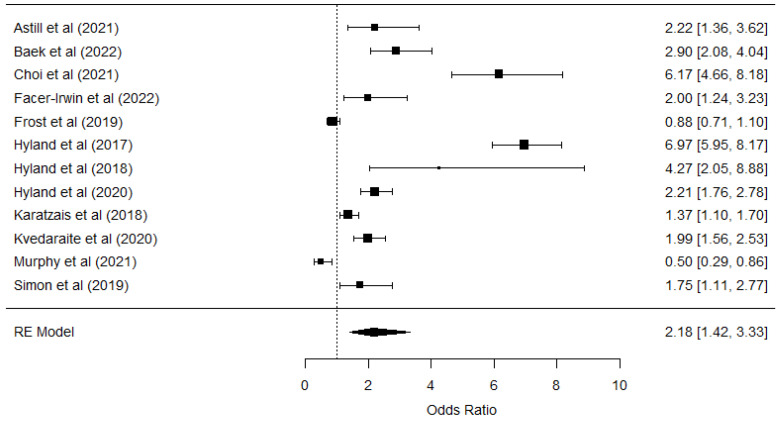
Forest plot of the unemployment and CPTSD relationship [17,40,42,47,51,53,54,55,56,58,61,66].

**Table 1 ijerph-22-00696-t001:** Database search syntax.

Scopus	TITLE (“complex PTSD” OR “complex posttraumatic stress disorder” OR “CPTSD” OR “PTSD” OR “posttraumatic stress disorder”) AND TITLE (“risk factors” OR “predictors” OR “unemployment”).
Web of Sciences	TI = ((“complex PTSD” OR “complex posttraumatic stress disorder” OR “CPTSD” OR “PTSD” OR “posttraumatic stress disorder”) AND (“risk factors” OR “predictors” OR “unemployment”))
PubMed	((“complex PTSD”[Title] OR “complex posttraumatic stress disorder”[Title] OR “CPTSD”[Title] OR “PTSD”[Title] OR “posttraumatic stress disorder”[Title]) AND (“risk factors”[Title] OR “predictors”[Title] OR “unemployment”[Title]))

**Table 2 ijerph-22-00696-t002:** Characteristics of the included studies with unemployment as a risk factor for PTSD and CPTSD.

ID	Authors	Design	Sample Type	N	% Female	Age (M ± SD)	N Unemp	Identification of PTSD/CPTSD
01	Ali et al. (2012) [39]	Cross-sectional	Survivors	300	39.3%	37.8 ± 14.0	234	Davidson Trauma Scale
02	Wright et al. (2021) [40]	Cross-sectional	General	222	-	16	137	International Trauma Questionnaire (ITQ)
03	Ayazi et al. (2012) [41]	Cross-sectional	General	1200	44%	-	111	Harvard Trauma Questionnaire (HTQ)
04	Baek et al. (2022) [42]	Cross-sectional	Defectors	503	84.5%	46 ± 13.2	-	International Trauma Questionnaire (ITQ)
05	Bronner et al. (2010) [43]	Longuitudinal	Clinical	588	56.8%	36.5 ± 7.0	150	Self-Rating Scale for PTSD (SRS-PTSD)
06	Cénat et al. (2014) [44]	Cross-sectional	General	1355	48.7%	31.57	221	Impact of Event Scale—Revised (IES-R)
07	Cerdá et al. (2013) [45]	Cross-sectional	General	1315	71.1%	-	-	Lifetime Violent Traumatic Event Experience
08	Cheng et al. (2015) [46]	Cross-sectional	Survivors	182	65.2%	-	88	Structured Clinical Interview for DSM-IV Axis I Disorders
09	Choi et al. (2021) [47]	Cross-sectional	General	800	-	-	-	International Trauma Questionnaire (ITQ)
10	Cohen et al. (2009) [48]	Cross-sectional	Clinical	850	100%	36.4 ± 8.3	637	Harvard Trauma Questionnairem (HTQ)
11	Cofini er al. (2015) [49]	Cross-sectional	Survivors	281	54%	43	103	Davidson Trauma Scale (DTS)
12	Eşsizoğlu et al. (2017) [50]	Cross-sectional	General	93	34.4%	28.28 ± 9.8	63	Traumatic Stress Symptom Scale (TSSC)
13	Facer-Irwin et al. (2022) [51]	Cross-sectional	Imprisoned	221	0%	31.3 ± 9.0	116	International Trauma Questionnaire (ITQ)
14	Frost et al. (2019) [17]	Cross-sectional	General	1051	68.4%	47.18	-	Life Events Checklist (LEC)
15	Gros et al. (2013) [52]	Cross-sectional	Veterans	92	6.5%	33.2 ± 9.0	48	Clinician Administered PTSD Scale
16	Hyland et al. (2017) [53]	Cross-sectional	General	2591	54.6%	24	-	International Trauma Questionnaire (ITQ)
17	Hyland et al. (2018) [54]	Cross-sectional	Refugees	110	51%	-	-	International Trauma Questionnaire (ITQ)
18	Hyland et al. (2021) [55]	Cross-sectional	General	1020	51%	43.1 ± 15.1	-	International Trauma Questionnaire (ITQ)
19	Karatzias et al. (2019) [56]	Cross-sectional	General	1051	68.4%	47.18 ± 15	-	International Trauma Questionnaire (ITQ)
20	Kimerling et al. (2009) [57]	Cross-sectional	General	6698	100%	-	-	Harvard Trauma Questionnaire (HTQ)
21	Kvedaraite et al. (2022) [58]	Cross-sectional	General	885	63.4%	37.96 ± 14.67	205	International Trauma Questionnaire (ITQ)
22	Lee et al. (2009) [59]	Cross-sectional	General	196	-	-	-	Schedules for Clinical Assessment in Neuropsychiatry
23	Mills et al. (2016) [60]	Longitudinal	Clinical	103	60%	33.4 ± 7.40	79	Clinician-Administered PTSD Scale
24	Murphy et al. (2021) [61]	Cross-sectional	Veterans	177	4.90%	-	127	International Trauma Questionnaire (ITQ)
25	Nandi et al. (2004) [62]	Longitudinal	General	1939	54.3%	-	-	Structured Clinical Interview for DSM-IV Axis I Disorders
26	Pazderka et al. (2022) [63]	Cross-sectional	General	159	100%	-	-	Lifetime Violent Traumatic Event Experience
27	Powers et al. (2014) [64]	Longitudinal	Clinical	327	36%	46 ± 18.00	141	Primary Care Post-Traumatic Stress Disorder Screen
28	Rybojad et al. (2016) [65]	Cross-sectional	Paramedics	100	14%	33.6 ± 9.30	-	Impact of Event Scale—Revised (IES-R)
29	Serrano et al. (2021) [21]	Cross-sectional	General	26,213	68.7%	49.5 ± 17.0	-	Davidson Trauma Scale (DTS)
30	Simon et al. (2019) [66]	Cross-sectional	Clinical	246	50%	47.37 ± 12	235	International Trauma Questionnaire (ITQ)
31	Stuber et al. (2010) [67]	Cross-sectional	Childhood	6542	52.3%	31.85 ± 7.5	1439	Posttraumatic Stress Diagnostic Scale
32	Teramoto et al. (2015) [68]	Cross-sectional	General	296	57.1%	57.10	-	Kessler Psychological Distress Scale
33	Weiss et al. (2011) [69]	Cross-sectional	Clinical	245	69.6%	43.7 ± 10.9	-	Clinician Administered PTSD Scale

**Table 3 ijerph-22-00696-t003:** Statistics used for meta-analysis and meta-regression for each included study.

ID	Author	PTSD			CPTSD			Region	GINI	NPIB	Quality Score
		OR	Log	VarLog	OR	Log	VarLog				
01	Ali et al. (2012) [39]	3.72	1.31370	0.0497	-		-	Pakistan	29.6	1588	6.0
02	Wright et al. (2021) [40]	1.58	0.45742	0.1555	2.22	0.79750	0.06235	United Kingdom	32.6	46,125	6.0
03	Ayazi et al. (2012) [41]	4.25	1.44690	0.0127	-	-	-	South Sudan	44.1	1071	7.0
04	Baek et al. (2022) [42]	1.30	0.26236	0.0263	2.90	1.06471	0.02847	South Korea	31.4	32,422	6.0
05	Bronner et al. (2010) [43]	2.56	0.94001	0.0951	-	-	-	Netherlands	26.0	57,025	10
06	Cénat et al. (2014) [44]	0.84	−0.17435	0.0007	-	-	-	Haiti	41.1	1742	6.0
07	Cerdá et al. (2013) [45]	1.44	0.36464	0.0100	-	-	-	Haiti	41.1	1742	7.0
08	Cheng et al. (2015) [46]	0.55	−0.59784	0.0746	-	-	-	China	37.1	12,720	7.0
09	Choi et al. (2021) [47]	-	-	-	6.17	1.81969	0.02061	South Korea	31.4	32,422	5.0
10	Cohen et al. (2009) [48]	1.00	0.00000	0.0154	-	-	-	Rwanda	43.7	966	7.0
11	Cofini er al (2015) [49]	2.10	0.74194	0.0489	-	-	-	Italy	35.2	34,776	8.0
12	Eşsizoğlu et al. (2017) [50]	1.47	0.38526	0.3908	-	-	-	Turkey	41.9	10,674	7.0
13	Facer-Irwin et al. (2022) [51]	1.90	0.64185	0.0309	2.00	0.69314	0.05981	United Kingdom	32.6	46,125	7.0
14	Frost et al. (2019) [17]	1.48	0.39204	0.1264	0.88	−0.12783	0.01258	United Kingdom	32.6	46,125	5.0
15	Gros et al. (2013) [52]	1.87	0.62594	0.1489	-	-	-	United States	39.8	76,329	9.0
16	Hyland et al. (2017) [53]	1.66	0.50682	0.0045	6.99	1.94448	0.00654	Denmark	27.5	67,790	7.5
17	Hyland et al. (2018) [54]	4.96	1.60141	0.1442	4.27	1.46161	0.13962	Lebanon	31.8	4136	6.5
18	Hyland et al. (2021) [55]	1.20	0.18232	0.0110	2.21	0.79299	0.01353	Ireland	31.8	103,983	6.5
19	Karatzias et al. (2019) [56]	0.98	−0.02020	0.0120	1.37	0.31481	0.01259	United Kingdom	32.6	46,125	8.0
20	Kimerling et al. (2009) [57]	1.60	0.47000	0.0019	-	-	-	United States	39.8	76,329	5.0
21	Kvedaraite et al. (2022) [58]	0.89	−0.11653	0.1226	1.99	0.68813	0.01542	Lithuanian	36	25,064	7.0
22	Lee et al. (2009) [61] [59]	1.60	0.47000	0.0051	-	-	-	Taiwan	34.2	35,513	7.0
23	Mills et al. (2016) [60]	1.74	0.55389	0.1320	-	-	-	Australia	34.3	52,084	7.0
24	Murphy et al. (2021) [61]	0.25	−1.38629	0.0085	0.50	−0.69314	0.07749	United Kingdom	32.6	46,125	7.5
25	Nandi et al. (2004) [62]	1.00	0.00000	0.0067	-	-	-	United States	39.8	76,329	9.0
26	Pazderka et al. (2022) [63]	5.18	1.64481	0.1004	-	-	-	Canada	31.7	54,917	6.0
27	Powers et al. (2014) [64]	1.03	0.02956	0.0403	-	-	-	United States	39.8	76,329	7.0
28	Rybojad et al. (2016) [65]	3.27	1.18479	0.2235	-	-	-	Poland	28.8	18,688	7.0
29	Serrano et al. (2021) [21]	1.18	0.16551	0.0005	-	-	-	Chile	44.9	15,355	7.0
30	Simon et al. (2019) [66]	-	-	-	1.75	0.55961	0.05499	United Kingdom	32.6	46,125	8,0
31	Stuber et al. (2010) [67]	2.01	0.69813	0.0232	-	-	-	United States	39.8	76,329	6.0
32	Teramoto et al. (2015) [68]	1.23	0.20701	0.0061	-	-	-	Japan	32.9	33,950	6.5
33	Weiss et al. (2011) [69]	1.57	0.45108	0.0547	-	-	-	United States	39.8	76,329	5.0

Abbreviations: PTSD = post-traumatic stress disorder; CPTSD = complex post-traumatic stress disorder; GINI = GINI coefficient; Nominal GDP = nominal gross domestic product; OR = odds ratio; LogOR = natural log of odds ratio; VarLog = variance of log odds ratio. **Note:** The GINI coefficient and nominal GDP (NGDP) values were obtained from the World Bank Data Portal (https://data.worldbank.org/) (accessed on 26 March 2025) using the most recent year available for each country (e.g., 2018, 2019, or 2020), which varies by nation depending on the latest data updates.

**Table 4 ijerph-22-00696-t004:** Results of the meta-analysis of unemployment as a risk factor for PTSD and CPTSD.

	REM									Heterogeneity Tests			Publication Bias	
	*k*	*n*	logOR	OR	*p*	PI_lower_	PI_upper_	IC_lower_	IC_upper_	Q	*p*	I^2^	ZEgger	*p*
PTSD	31	56,905	0.3826	1.500	0.000	0.457	4.702	0.165	0.600	804.155	0.001	98.02%	2.0813	0.037
CPTSD	12	8047	0.743	2.180	0.000	0.501	8.808	0.346	1.138	405.014	0.001	96.42%	−0.2670	0.789

Abbreviations: *k* = number of included studies; *n* = Sample size of included studies; PTSD = Post-traumatic stress disorder; CPTSD = complex post-traumatic stress disorder; OR = odds ratio; CI = confidence interval; Q = Cochran’s test; I^2^ = heterogeneity index; ZEgger = Egger’s regression.

**Table 5 ijerph-22-00696-t005:** Meta-regression for unemployment, PTSD and CPTSD.

								Heterogeneity Tests		
	*k*	Mod	Estimate	SE	*p*	IC_lower_	IC_upper_	Q	*p*	I^2^
PTSD	31	GINI	−0.0242	0.0221	0.2733	−0.0659	0.0224	799.9109	0.001	97.76%
		NGDP	−0.0000	0.0000	0.4926	−0.0000	0.0000	750.4517	0.001	97.83%
CPTSD	12	GINI	−0.2125	0.1014	0.0649	−0.4113	−0.0136	223.8095	0.001	94.94%
		NGDP	−0.0000	0.0000	0.8294	−0.0000	0.0000	345.4163	0.001	96.32%

Abbreviations: *k* = number of included studies; PTSD = post-traumatic stress disorder; CPTSD = complex post-traumatic stress disorder; Estimate = estimation of the effect of the variable; SE = standard error of estimate; CI = confidence interval; I^2^ = heterogeneity index; Q = Cochran’s test.

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
