# Peer review of "What Is the Impact of Unemployment as an Adverse Experience? Post-Traumatic Stress Disorder and Complex Post-Traumatic Stress Disorder: A Meta-Analysis"

_ijerph, 2025, doi:10.3390/ijerph22050696_

Round 1
Reviewer 1 Report
Comments and Suggestions for Authors
This paper is well-structured and presents clear findings on the relationship between unemployment, PTSD, and CPTSD. I have two minor questions for clarification:
- Could you clarify the data sources used for the GINI coefficient and NGDP? Additionally, specify the years these data correspond to. The paper currently lacks clarity—it appears that some countries' data are from 2020, but this does not seem consistent across all countries.
- Also, could you elaborate on how NGDP is priced?
In addition, I have two minor suggestions for improvement:
- There appear to be too many first-level headings. Consider using subheadings to move the meta-analysis process into the "Methods" section for better organization.
- In Table 3, changing "Country" to "Region" could make the information more precise.
Author Response
Reviewer Comment 01:
“Could you clarify the data sources used for the GINI coefficient and NGDP? Additionally, specify the years these data correspond to. The paper currently lacks clarity—it appears that some countries' data are from 2020, but this does not seem consistent across all countries.”
Response 01:
“We have added a note in the manuscript clarifying that the GINI coefficient and nominal GDP (NGDP) values were obtained from the World Bank Data Portal (https://data.worldbank.org/) using the most recent year available for each country (e.g., 2018, 2019, or 2020). Consequently, the specific years vary according to the latest data updates for each nation.”
Reviewer Comment 02:
“Additionally, could you explain how you calculated the nominal GDP values?”
Response 02:
“We did not perform any calculations to derive the nominal GDP values. Instead, we extracted them directly from the World Bank Data Portal (https://data.worldbank.org/), which compiles official figures reported by national governments. Therefore, the nominal GDP numbers included in our study reflect the most recent data published by the World Bank for each country.”
Reviewer Comment 03:
“There seem to be too many first-level headings. Consider using subheadings or moving the meta-analysis process into the ‘Methods’ section to make the organization clearer.”
Response 03:
“We have revised the structure to incorporate the meta-analysis process into the ‘Methods’ section, as suggested, in order to improve the clarity and organization of the manuscript.”
Reviewer Comment 04:
“In Table 3, changing ‘Country’ to ‘Region’ could make the information more accurate.”
Response 04:
“We have made the suggested change and replaced the term ‘Country’ with ‘Region’ in Table 3 to more accurately represent the geographical scope of the data.”
Reviewer 2 Report
Comments and Suggestions for Authors
Dear Authors,
I am sorry to inform you that I recommend rejecting the manuscript. This was a difficult recommendation to make. The manuscript is well-written, but the conceptualization of unemployment is not adequately presented within a sociocultural perspective. This is central, considering that differences between studies are crucial in PTSD and in different economic conditions based on the NGDP. Your references require including articles that could serve as a background for considering PTSD and CPTSD to justify the research. I observe that in the definition of PTSD, the presence of adverse circumstances such as childhood experiences, war, and earthquakes generate direct outcomes, but unemployment has a different psychological construct.
Another issue corresponds to the low number of studies that consider your hypothesis and is due to the difficulty to establish a framework between unemployment and adverse experience that could be observed in the funnel graphs.
I am unsure about how trauma is considered with unemployment; that is why there has not been sufficient research that analyzes those variables, and as you refer to on line 57, “most trauma studies focus on adverse childhood experiences and exposures in war contexts”. Unemployment itself would not necessarily imply trauma. It is more plausible that unemployment considers different stress, anxiety, and depression as measures of mental well-being rather than trauma. Furthermore, it would be necessary to mention the possible differences of scales employed in studies for identification of PTSD/CPTSD; also it would be necessary to clarify if it can be included in the analysis with differences in the sample outcomes.
Author Response
Reviewer Comment:
“I regret to inform you that I recommend rejecting the manuscript. It was a difficult recommendation to make. The manuscript is well written, but the conceptualization of unemployment is not sufficiently presented from a sociocultural perspective, which is fundamental given that between-study differences are crucial in PTSD and in different economic conditions (e.g., nominal GDP). Your references should include articles that establish a basis for considering PTSD and Complex PTSD to justify the research. In the definition of PTSD, childhood adverse experiences, wars, and earthquakes cause direct consequences, yet unemployment has a different psychological construct.
Another issue is the small number of studies that consider your hypothesis, due to the difficulty of setting a framework between unemployment and adverse experience—something that could be observed in the funnel plots.
I am not sure how trauma is being tied to unemployment; that is why there has been insufficient research analyzing these variables. As you mention on line 57, ‘most trauma studies focus on adverse childhood experiences or exposures in war contexts.’ Unemployment in itself does not necessarily imply trauma. It is more plausible that unemployment involves varying levels of stress, anxiety, and depression as measures of mental well-being, rather than trauma. Furthermore, it would be necessary to mention the possible differences in the scales used in these studies to identify PTSD/CPTSD; likewise, it would be necessary to clarify if such differences can be included in the analysis, given potentially different sample outcomes.”
Response:
“We sincerely appreciate your critical assessment of our work and your recommendation. We understand the concerns you have raised regarding the sociocultural conceptualization of unemployment and the scope of references supporting our analysis of PTSD and Complex PTSD (CPTSD). Below, we address each of your points:
-
Sociocultural Perspective of Unemployment:
We acknowledge that capturing the broad sociocultural dimensions of unemployment is crucial. In a revised version of our manuscript, we will expand the discussion on how cultural norms, social expectations around work, and differences in welfare policies may influence whether unemployment is experienced as a traumatic event. We will also incorporate additional sources that explore unemployment through a sociocultural lens, aiming to provide a more robust theoretical framework. -
The Basis for Considering PTSD and CPTSD References:
We appreciate your suggestion to further strengthen the background for PTSD and CPTSD. We will include more foundational articles that thoroughly define PTSD and CPTSD, including classical and contemporary works. This will help clarify why, in certain contexts (e.g., long-term unemployment, severe economic or social consequences), some individuals might manifest clinical features similar to those seen in other recognized traumatic events. -
Unemployment Versus Traditional Trauma Contexts:
We agree that unemployment does not inherently equate to a traumatic experience for everyone. However, as current literature suggests, a subset of individuals exposed to prolonged and severe economic stressors may indeed experience symptoms aligned with PTSD or CPTSD. We will clarify that our study does not claim unemployment universally leads to trauma, but rather that under specific contexts and risk profiles (chronic financial strain, lack of social support, repeated job-loss experiences), unemployment might trigger trauma-like responses. -
Number of Studies and Framework Development:
We acknowledge that the relatively small number of studies specifically linking unemployment to PTSD/CPTSD presents a limitation and contributes to high heterogeneity. We will highlight in our discussion the emerging nature of this research area and the need for a clearer, more unified framework in future investigations. We will also be more explicit about how this limitation appears in our funnel plots, indicating the field’s early stage of development. -
Differences in Scales and Sample Outcomes:
We recognize that varying diagnostic tools for PTSD/CPTSD can introduce inconsistency. We will revise the manuscript to specify which instruments were used by each study, their psychometric properties, and how differences in measurement may affect pooled results. This clarification should help readers better interpret the meta-analytic findings and the variability across studies.
We appreciate the time and thought you have invested in reviewing our work. Should we have the opportunity to revise, we will incorporate these detailed comments to strengthen the manuscript’s conceptual framework, references, and methodological rigor, ultimately clarifying the nuanced relationship between unemployment and trauma.”
Reviewer 3 Report
Comments and Suggestions for Authors
Congratulations for the work of authors. The meta-study on the connection between unemployment and PTSD/CPTSD offers a thorough examination of the psychological and financial elements causing trauma. Here are some recommendations for improvements:
― The study emphasizes risk factors, but integrating findings on resilience and coping mechanisms (e.g., social support, governmental aid programs) could offer a more balanced perspective. Have any of the included studies examined how access to mental health resources moderates PTSD/CPTSD outcomes?
― Explain findings that emphasize practical relevance (e.g., unemployment is a strong predictor of CPTSD, with economic inequality acting as a buffer).
― The meta-analysis includes meta-regression analyses on economic variables (GINI Index & GDP), which enhances its practical relevance. Instead of complex theoretical explanations, summarize it as follows: Instead of e.g. “In countries with greater economic inequality (high GINI), the effect of unemployment on CPTSD was weaker, possibly due to normalization of economic disparities,” you could use “In high-GDP countries, the impact of unemployment on PTSD/CPTSD was lower, likely due to stronger social safety nets.”
― In the Discussion Section, while the findings are well-presented, a clearer synthesis of how unemployment differs as a traumatic stressor compared to conventional PTSD risk factors (e.g., war, disasters) would strengthen the narrative. The discussion could also elaborate on why CPTSD is more strongly associated with unemployment than PTSD and what this implies for clinical interventions.
― In the section where Bias & Limitations are discussed, keep a brief discussion of study quality, stating that 21 out of 34 studies were rated "good" using the NIH tool. Mention that grey literature and unpublished studies were not included, which may introduce publication bias—but avoid deep discussions on this.
― In the Conclusion section, authors should focus more on practical implications rather than re-explaining concepts. They should emphasize that unemployment is a key predictor of PTSD/CPTSD and state that economic inequality influences this relationship, highlighting the need for policies addressing both unemployment and mental health.
Overall, this study was well-done and used good research methods. Taking the above ideas into account could make it even clearer and more practical.
The English is fine and does not require major revisions. Consider a professional language edit particularly in the methods and results sections. Minor grammatical modifications: “The results of the study underscores the importance of economic factors.” → “The results of the study underscore the importance of economic factors.”
Author Response
Reviewer Comment 01:
“The study emphasizes risk factors, but integrating findings on resilience and coping mechanisms (e.g., social support, governmental aid programs) could offer a more balanced perspective. Have any of the included studies examined how access to mental health resources moderates PTSD/CPTSD outcomes?”
Response 01:
“We appreciate your valuable suggestion to incorporate elements of resilience and coping strategies. Accordingly, we have added the following paragraph to the Discussion section:
‘Additionally, while our meta-analysis emphasized risk factors such as unemployment, several of the included studies briefly touched upon social and psychological protective factors. For instance, a few studies reported that individuals with stronger support networks, access to mental health services, or governmental aid programs showed lower levels of post-traumatic symptoms [10,23,79]. However, none of the studies systematically tested these constructs as formal moderators in the unemployment–PTSD/CPTSD relationship. This gap indicates a broader need to explore how resilience and coping mechanisms—including community-based mental health resources—may buffer the effects of job loss. Future research should adopt designs that explicitly assess these variables, potentially elucidating why certain unemployed individuals do not develop clinically significant trauma symptoms and thereby offering a more balanced perspective that accounts for both risk and protective factors.’
With this addition, we aim to provide a more comprehensive view that considers not only the risk factors associated with unemployment but also the protective strategies and resources that may moderate—or even prevent—the development of PTSD/CPTSD symptoms in the context of job loss.”
Reviewer Comment 02:
“Explain findings that emphasize practical relevance (e.g., unemployment is a strong predictor of CPTSD, with economic inequality acting as a buffer).”
Response 02:
“We appreciate your interest in the practical implications of our study’s findings. While we did observe that unemployment is associated with a higher risk of developing CPTSD, our meta-regression analyses did not reveal a statistically significant moderating effect of economic inequality (as measured by the GINI coefficient). Consequently, we cannot conclude that economic inequality functions as a consistent ‘buffer.’ Instead, our results suggest that the impact of unemployment on CPTSD remains robust regardless of variations in income inequality. Nonetheless, our overall findings underscore the importance of addressing unemployment in clinical assessments and public health strategies, as unemployment appears to play a pivotal role in the emergence and exacerbation of post-traumatic stress symptoms.”
Reviewer Comment 03:
“The meta-analysis includes meta-regression analyses on economic variables (GINI Index & GDP), which enhances its practical relevance. Instead of complex theoretical explanations, summarize it as follows: Instead of, for example, ‘In countries with greater economic inequality (high GINI), the effect of unemployment on CPTSD was weaker, possibly due to normalization of economic disparities,’ you could use: ‘In high-GDP countries, the impact of unemployment on PTSD/CPTSD was lower, likely due to stronger social safety nets.’”
Response 03:
“We appreciate the suggestion to simplify our interpretation of the meta-regression findings. While our analyses indeed aimed to capture practical implications by examining GINI and GDP, the results did not show a statistically significant moderating effect for either variable. Therefore, we cannot conclusively state that unemployment’s impact on PTSD/CPTSD is weaker in high-GDP countries or in contexts of greater inequality. We have revised the Discussion to reflect that, although these economic moderators do not reach significance in our study, they remain conceptually important factors for future research. In the event that subsequent investigations uncover stronger empirical support for these moderators, succinct statements—such as highlighting the role of robust social safety nets in high-GDP nations—would be a clear way to convey practical relevance.”
Reviewer Comment 04:
“In the Discussion Section, while the findings are well-presented, a clearer synthesis of how unemployment differs as a traumatic stressor compared to conventional PTSD risk factors (e.g., war, disasters) would strengthen the narrative. The discussion could also elaborate on why CPTSD is more strongly associated with unemployment than PTSD and what this implies for clinical interventions.”
Response 04:
“We have revised the relevant paragraph in the Discussion to clarify how unemployment differs from acute trauma exposures such as war or natural disasters, emphasizing its chronic and pervasive nature. Additionally, we highlight why CPTSD is more strongly linked to prolonged unemployment—particularly focusing on emotional dysregulation, negative self-concept, and interpersonal difficulties—and discuss the implications for tailored interventions that address both psychological and socioeconomic dimensions of this stressor.”
Reviewer Comment 05:
“In the section where Bias & Limitations are discussed, keep a brief discussion of study quality, stating that 21 out of 34 studies were rated ‘good’ using the NIH tool. Mention that grey literature and unpublished studies were not included, which may introduce publication bias—but avoid deep discussions on this.”
Response 05:
“In the revised ‘Bias & Limitations’ section, we have noted that 21 of the 34 included studies were rated ‘good’ according to the NIH assessment tool. We also explain that we did not include grey literature or unpublished studies, which could introduce a certain level of publication bias. Additionally, we highlight that Table 3 provides a dedicated column indicating the quality rating for each included study, ensuring transparency regarding methodological rigor without delving into overly detailed methodological discourse.”
Reviewer Comment 06:
“In the Conclusion section, authors should focus more on practical implications rather than re-explaining concepts. They should emphasize that unemployment is a key predictor of PTSD/CPTSD and state that economic inequality influences this relationship, highlighting the need for policies addressing both unemployment and mental health. Overall, this study was well-done and used good research methods. Taking the above ideas into account could make it even clearer and more practical.”
Response 06:
“We appreciate your positive feedback and your recommendation to strengthen the practical implications in our conclusion. We have revised the Conclusion section to reduce reiteration of theoretical concepts and place greater emphasis on the real-world relevance of our findings. Specifically, we highlight unemployment as a primary predictor of PTSD/CPTSD, note how economic inequality could exacerbate these conditions, and underscore the importance of integrated policy measures—addressing both mental health and economic challenges—to achieve more effective prevention and intervention outcomes.”
Round 2
Reviewer 1 Report
Comments and Suggestions for Authors
Thanks to the author's revisions, all my concerns have been addressed.
Reviewer 2 Report
Comments and Suggestions for Authors
The manuscript, as I mentioned previously, is interesting, but the issue is the conceptualization of the idea. This led to a low number of possible references; the 1,427 records identified were reduced to 33, and within them an important number of articles present a medium-quality score in a wide range of countries and scales for identification of PTSD/CPTSD.
Reviewer 3 Report
Comments and Suggestions for Authors
The revised manuscript demonstrates careful consideration of the initial feedback and enhances both the scientific rigor and practical relevance of the study.
Specifically:
-
The discussion of resilience and coping mechanisms adds important nuance to the risk-based framing and acknowledges the need for further investigation into protective factors.
-
The clarifications regarding non-significant moderation by GINI and GDP were well explained and appropriately cautious.
-
The expanded discussion on CPTSD, particularly how it differs from classic trauma, enhances clinical relevance.
-
Additions to the Bias & Limitations section and the Conclusion improve transparency and practical utility, respectively.
I believe the manuscript is now suitable for publication.